# Early-life experience reorganizes neuromodulatory regulation of stage-specific behavioral responses and individuality dimensions during development

**Reemy Ali Nasser, Yuval Harel, Shay Stern***

Faculty of Biology, Technion - Israel Institute of Technology, Haifa, Israel

**Abstract** Early-life experiences may promote stereotyped behavioral alterations that are dynamic across development time, but also behavioral responses that are variable among individuals, even when initially exposed to the same stimulus. Here, by utilizing longitudinal monitoring of *Caenorhabditis elegans* individuals throughout development we show that behavioral effects of early-life starvation are exposed during early and late developmental stages and buffered during intermediate stages of development. We further found that both dopamine and serotonin shape the discontinuous behavioral responses by opposite and temporally segregated functions across development time. While dopamine buffers behavioral responses during intermediate developmental stages, serotonin promotes behavioral sensitivity to stress during early and late stages. Interestingly, unsupervised analysis of individual biases across development uncovered multiple individuality dimensions that coexist within stressed and unstressed populations and further identified experience-dependent effects on variation within specific individuality dimensions. These results provide insight into the complex temporal regulation of behavioral plasticity across developmental timescales, structuring shared and unique individual responses to early-life experiences.

**\*For correspondence:**
sstern@technion.ac.il

**Competing interest:** The authors declare that no competing interests exist.

## Editor's evaluation

Early life stress can have profound effects on animal behavior, including potential influences on individuality. Here, the authors use a rich new dataset to convincingly demonstrate that the behavioral consequences of early life stress in *C. elegans* can be buffered by neuromodulators previously implicated in patterns of individuality. While much remains to be learned about the mechanisms by which stress might influence individuality, these studies report important advances that will be of interest to neurobiologists studying interactions between behavior, neuromodulation, stress, and individuality.

## Introduction

Long-term behavioral patterns across development are highly dynamic across and within developmental stages and are temporally synchronized with the individual's developmental clock. For instance, flies show differences in foraging behavior that depend on their larval stage (*Sokolowski et al., 1984*), fish internally modify their startle response across life (*Kimmel et al., 1974*) and stomatogastric motor patterns are regulated across development (*Rehm et al., 2008*). In addition, fear-extinction learning is inhibited during adolescence compared to other life stages in humans and mice (*Pattwell et al.,*

*2012*). Long-term behavioral outputs are influenced by the changes in the internal state of individuals, as well as by their past and current environmental exposures. In particular, animals may be transiently exposed to environmental perturbations at different stages of life, but experiences during early developmental windows, which are usually referred to as critical or sensitive periods, were shown to generate long-lasting effects (*Lorenz, 1935*; *Korosi et al., 2012*; *Jin et al., 2016*; *Nevitt et al., 1994*; *Remy and Hobert, 2005*). This stable imprinting of early memories has the potential to increase survival and reproduction during later life stages of the organism (*Immelmann, 1975*). However, a complete temporal view of the long-term effects of early experiences on behavior throughout development, across and within all stages, is still lacking.

While long-lasting effects on behavior may be shared by many individuals, reflected by stereotypic behavioral responses following an early-life experience, individuals within the same population may also show unique patterns of long-term behavior that distinguish them from each other. This inter-individual variation in behavioral responses may be exposed even when animals are initially experiencing the same early conditions. Here we study how early-life experiences shape stage-specific behavioral patterns across development and how they affect the diversity in long-term behavioral responses among individuals. Consistent behavioral individuality within isogenic populations that were raised in the same environment has been previously described in various species, including in the pea aphid (*Schuett et al., 2011*), *Drosophila melanogaster* (*Buchanan et al., 2015*; *Kain et al., 2012*; *Linneweber et al., 2020*), clonal fish (*Bierbach et al., 2017*), and mice (*Freund et al., 2013*). The nematode *Caenorhabditis elegans* is an ideal system to study how early-life experiences shape long-term behavior and inter-individual variation across developmental timescales due to their short development time of 2.5 days and the homogeneous populations generated by the self-fertilizing reproduction mode of the hermaphrodite. It was previously shown that under normal growth conditions, *C. elegans* shows both stereotypic patterns of long-term behavior and consistent individual biases within the isogenic population (*Stern et al., 2017*).

By continuously tracking the locomotory behavior of single individuals following transient periods of starvation early in life throughout their complete developmental trajectory, we show that early-life starvation exposes long-term behavioral plasticity that is discontinuous over development time. Temporal differences in long-term behavioral responses to early stress are reflected by strong behavioral modifications during early and late developmental stages and the buffering of behavioral effects during intermediate stages of development. We further found that dopamine maintains the buffering of behavioral responses during mid-development and serotonin promotes behavioral sensitivity to early starvation during early and late stages of development. Moreover, by performing unsupervised analysis of patterns of individual biases across development, we identified a spectrum of temporal individuality dimensions that are dominant within stressed and unstressed populations. Both the early-life history and neuromodulatory state of the population affect variation within specific individuality dimensions. These results show how a transient early-life environment shapes a long-term behavioral structure of stereotypic and variable responses across developmental stages.

## Results

### Early-life stress generates discontinuous and distinct behavioral effects at different stages of development

To study how stressful environments early in life influence long-term behavioral patterns across and within all developmental stages we continuously tracked the behavior of individual animals following a transient period of starvation early in development. Imaging was performed using a custom-made multi-camera imaging system across the full developmental trajectory of *C. elegans* individuals (55 hr), at high spatiotemporal resolution (3 fps, ~10 μm) and in a tightly controlled environment (*Stern et al., 2017*). First larval stage (L1) animals that hatch into an environment that completely lacks a food source do not grow and their development is arrested (*Greenwald and Horvitz, 1982*; *Johnson et al., 1984*; *Baugh, 2013*). Following L1 arrest, when animals encounter food, they resume their normal developmental trajectory to reach adulthood. This early-stress paradigm allows us to maintain a homogeneous stress environment across individuals at their earliest stage of development, immediately after hatching.

We continuously monitored single N2 wild-type individuals grown in isolation from their first larval stage to 16 hr of adulthood (*n* = 456) on defined concentrations of UV-killed OP50 bacteria, following periods of stress ranging from 1 to 4 days of early starvation (*Figure 1A, B*). In parallel, we tracked the behavior of individuals grown continuously on food, without experiencing starvation (*Figure 1A, B*). Animals exposed to early starvation required more time to complete their development (*Figure 1—figure supplement 1A*). To align developmental trajectories of different individuals in time, we age-normalized individuals by dividing each developmental stage, detected by the lethargus period during molting (*Cassada and Russell, 1975*), into 75 time windows (*Figure 1—figure supplement 1C*; *Stern et al., 2017*). Whilst growing in a food environment, *C. elegans* shifts between two behavioral states called roaming and dwelling that last seconds to minutes (*Ben Arous et al., 2009*; *Flavell et al., 2013*; *Fujiwara et al., 2002*; *Stern et al., 2017*). During a roaming episode animals explore a large area by high-speed forward movements, while in the dwelling episode they show dramatically less exploration due to low-speed movements coupled with frequent reorientations (*Figure 1—figure supplement 1B*). We quantified long-term patterns of locomotory behavior shown by individuals throughout development by measuring two behavioral parameters: fraction of time spent roaming, and speed during roaming episodes.

Unstressed individuals hatching in a food environment show dynamic behavioral structures of roaming activity across development, as was previously shown (*Stern et al., 2017*; *Figure 1B–D*). We found that a transient exposure to early-life starvation generates alterations in long-term behavioral patterns throughout development that were distinct and discontinuous across and within developmental stages. A short early-starvation period of 1 day strongly decreased average roaming activity levels during the L1 and adult stages compared to unstressed individuals (*Figure 1C, E*; *Figure 1—figure supplement 1D*). In contrast, while 1 day of early starvation modified within-stage temporal behavioral structures by shifting roaming activity peaks to later time windows during the L2 and L3 stages (*Figure 1C*; *Figure 1—figure supplement 1E*), average roaming activity was not decreased during these stages (*Figure 1E*). Similarly, we found that average roaming activity level was also maintained during the L4 stage, following 1-day starvation (*Figure 1C, E*; *Figure 1—figure supplement 1D*). Interestingly, animals exposed to longer starvation periods of 3 and 4 days further showed strong roaming decrease during L1 and adulthood, but exhibited only minor effects on average roaming activity within the intermediate L2, L3, and L4 larval stages (*Figure 1D, E*; *Figure 1—figure supplement 1D*). Furthermore, while the average size of animals that experienced early starvation was slightly decreased (~10%), a comparison of starved and unstarved individuals within the same size range (size-matched) showed similar stage-specific effects on roaming activity during L1 and adulthood following early starvation (*Figure 1—figure supplement 1F*). These results indicate that, while the memory of early starvation is maintained throughout development to expose strong decrease in average roaming behavior during L1 and adulthood, behavioral effects are buffered across intermediate development times.

Similar to the stage-specific effects of early starvation on the fraction of time spent roaming, instantaneous speed during roaming episodes in individuals exposed to early starvation was affected more strongly at the L1 and adult stages, compared to the intermediate stages (*Figure 1—figure supplement 1G*). In summary, transient early-life starvation discontinuously reshapes long-term behavioral patterns across development time by exposing strong behavioral alterations at early and late developmental stages and buffered effects during intermediate stages.

## Unsupervised analysis uncovers multiple individuality dimensions within stressed and unstressed populations

The longitudinal measurements in single animals across development allow us to further quantify long-term inter-individual diversity within stressed and unstressed populations. Following early starvation, different individuals show substantial variation in long-term behavioral responses. For instance, during L1 and adulthood, a fraction of wild-type individuals that were exposed to early stress show 8- to 10-fold decrease in roaming activity relative to the average roaming level of the unstressed population, while other stressed individuals show roaming activity which is indistinguishable from unstressed animals (*Figure 1—figure supplement 1D*). Behavioral individuality is usually defined as a consistent tendency of an individual to show the same behavioral bias relative to the population across long time-periods (*Bierbach et al., 2017*; *Buchanan et al., 2015*; *Kain et al., 2012*; *Stern et al., 2017*;

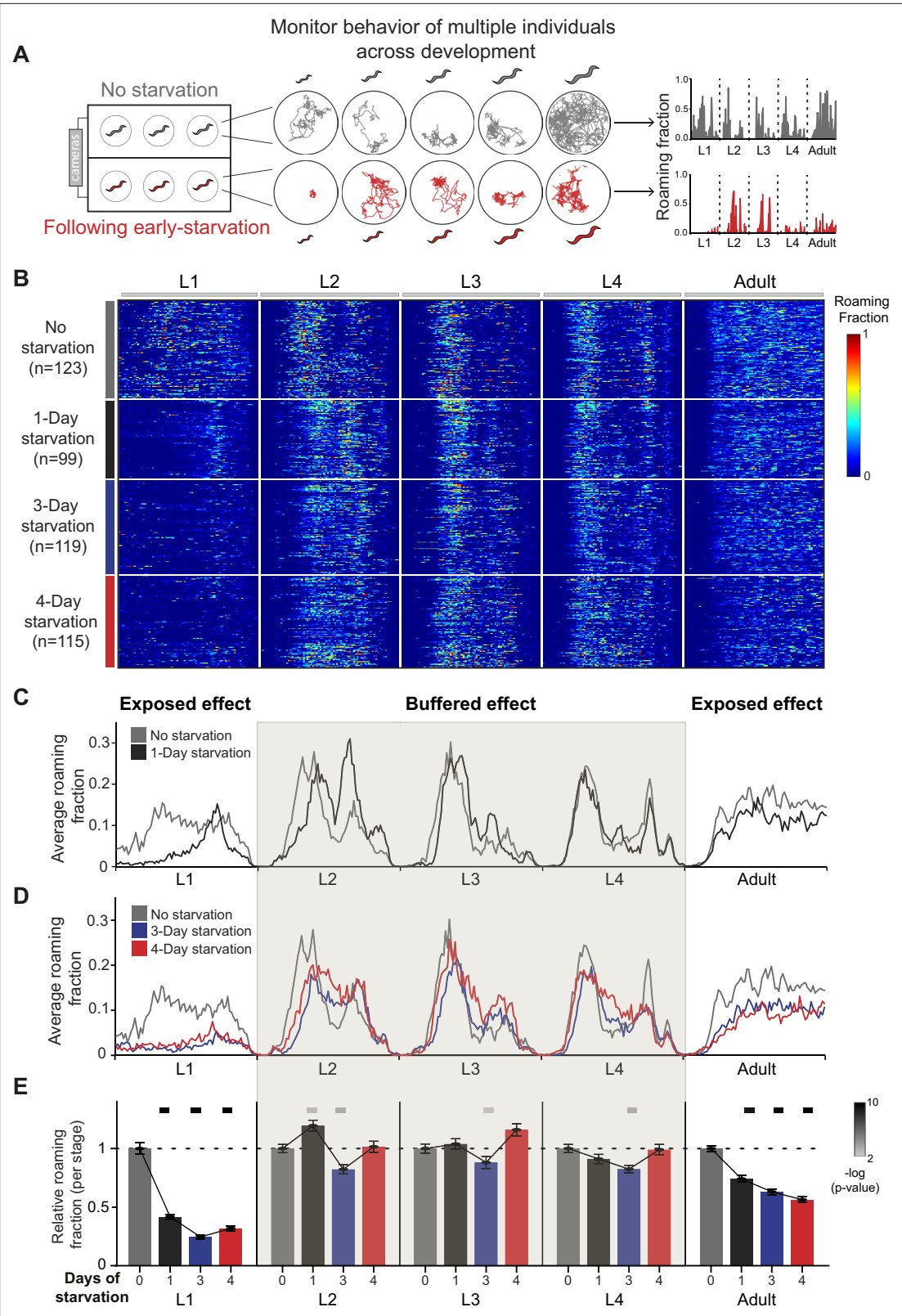

**Figure 1.** Long-term behavioral tracking of *C. elegans* following early starvation reveals discontinuous behavioral effects across developmental stages. (**A**) Multi-camera imaging system allows longitudinal behavioral tracking of multiple individual worms across all stages of development following early L1 starvation and without starvation, under tightly controlled environmental conditions. Shown are representative locomotion trajectories (middle) and age-normalized roaming activity (right) of post-starved (red) and well-fed (gray) individual worms across all four larval stages and adulthood. Normalization

*Figure 1 continued on next page*

*Figure 1 continued*

equally divides each stage into 75 time bins. (**B**) Roaming and dwelling behavior of wild-type N2 animals without early starvation (*n* = 123) and following 1-day (*n* = 99), 3-day (*n* = 119), and 4-day starvation (*n* = 115). Each row indicates the age-normalized behavior of one individual across all developmental stages. The different stages are separated by vertical white lines indicating the middle of the lethargus state. Color bar represents the fraction of time spent roaming in each of the 375 time bins. (**C**) Average roaming fraction of 1-day starved wild-type animals compared to the unstarved population. (**D**) Average roaming fraction of 3- and 4-day starved wild-type animals compared to the unstarved population. (**E**) Average roaming fraction relative to the unstarved population in each developmental stage. Shading highlights intermediate stages of development in which average behavioral effects within a stage are buffered. Error bars indicate standard error of the mean. Upper bars indicate statistical significance (Wilcoxon rank-sum test, False Discovery Rate (FDR) corrected) of difference in average roaming fraction between starved and unstarved populations ($-\log$(p-value), indicated are p-values <0.01).

The online version of this article includes the following figure supplement(s) for figure 1:

**Figure supplement 1.** Development time, behavioral trajectory synchronization, and roaming quantification in starved and unstarved wild-type individuals.

*Schuett et al., 2011*). However, individuals may also show alternative patterns of temporal behavioral biases relative to the population that are not random and represent more complex structures of individual biases over time. Here, we extend the 'classic' analysis of individuality and ask if alternative individuality dimensions coexist within *C. elegans* populations across development.

To analyze long-term individual biases in behavior we first systematically rank individuals based on their roaming activity compared to all other individuals within the same experiment across developmental windows (50 time bins) (*Figure 2A, B*). The rank approach allows us to homogeneously compare between individuals at each developmental window. To take an unsupervised approach for detecting temporal patterns of individual biases that are dominant within stressed and unstressed populations we performed principal component analysis (PCA) of the temporal behavioral ranks of all wild-type individuals (*n* = 456). Following PCA, each individual within the population is represented by its score (value) in each of the PC dimensions. We identified statistically significant PC dimensions by comparing the variances in PC scores within each dimension to those obtained from PCA of a randomly shuffled rank dataset (*Figure 2C*; *Figure 2—figure supplement 1A*) or to score variances of a shuffled rank dataset within the same PCA space (*Figure 2—figure supplement 1B*).

We found that among the significant PC dimensions (*Figure 2D–F*; *Figure 2—figure supplement 1F*), the three major PCs (PC1–3) captured three distinct dimensions of temporal individuality patterns within stressed and unstressed populations (*Figure 2D–F*). PC1, which explained the majority of temporal variation in individual biases over time had eigenvector components of the same sign, indicating an individuality dimension of animals that consistently roam more or less than the population homogeneously across all developmental stages (*Figure 2D*). The individuality dimension identified by PC1 unbiasedly recaptured a known mode of consistent individuality that was previously identified using a pre-defined index of long-term behavioral consistency across development (*Stern et al., 2017*; *Figure 2—figure supplement 1C, D*). This was further verified by the high correlation between the pre-defined consistency index and scores of PC1 across individuals (*R* = 0.9) (*Figure 2—figure supplement 1E*). Interestingly, other PC dimensions identified uncharacterized individuality patterns. PC2, which had opposite signs of eigenvector components before and after mid-development captured an individuality dimension that includes individuals that switch their behavioral bias once, during the L3 stage, from roaming more to roaming less than the population and vice versa (*Figure 2E*). In addition, PC3, which had signs of eigenvector components that switch twice during development (at the end of L1 and L4), identified individuals that show the same behavioral bias during L1 and adulthood, which is opposite to their behavioral bias during intermediate stages (*Figure 2F*). Other significant PC dimensions showed more complex dynamics of temporal individual biases across development, displaying multiple bias switching within developmental stages (*Figure 2—figure supplement 1F*). PC individual scores in these alternative PC dimensions did not correlate with the pre-defined consistency index (*R* = 0.003–0.09) (*Figure 2—figure supplement 1E*), further indicating that they indeed represent uncharacterized modes of temporal individuality.

Inter-individual variation in PC scores within a specific PC dimension reflects how extreme individuals are within a population toward the identified individuality dimension. We found that wild-type populations with different early-life experiences show extreme inter-individual variation in multiple PC dimensions, compared to a randomly shuffled rank dataset within the same PCA space or within a

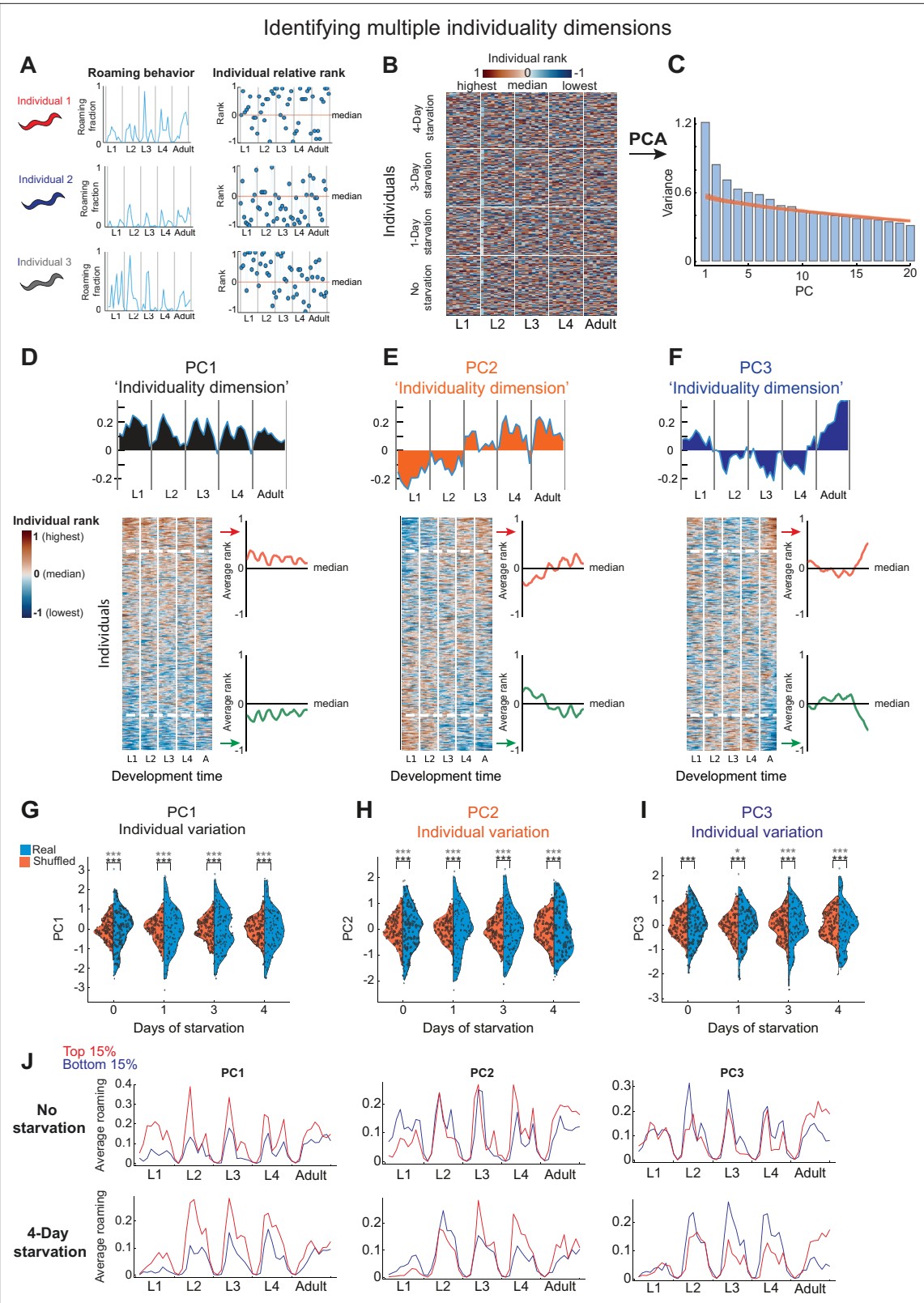

**Figure 2.** Unsupervised analysis of temporal individuality dimensions across development. (**A**) Individual animals are ranked based on their roaming activity in each time window compared to other individuals within the same experiment. (**B**) Heat-map represents relative rank of all N2 wild-type individuals ($n = 456$) across 50 time windows (10 per developmental stage). (**C**) Variance explained by each of the first 20 PCs following principal component analysis (PCA; blue bars), compared to the variance explained by the first 20 PCs extracted from PCA of a shuffled dataset (500 repetitions,

*Figure 2 continued on next page*

*Figure 2 continued*

orange lines). (**D–F**) PC1–3 represent the first three temporal individuality dimensions. For each PC individuality dimension shown are its components in each time window (top) and individuals sorted based on their PC score (bottom). Heat map is smoothed (4-bins window) for visual clarity. Average relative rank is plotted for extreme individuals (top and bottom 15%) within each individuality dimension. Midline represents the population median. (**G–I**) Distributions of individual scores (blue) within starved and unstarved wild-type populations for PC1–3 individuality dimensions, compared to distributions of individual scores of a shuffled dataset in the same PCA space (orange). p-values above distributions were calculated using bootstrapping (see Methods) for significance of difference in PC1–3 variation to variation of a shuffled dataset in the same PCA space (bottom asterisks) or in a PCA space extracted from the shuffled dataset (upper asterisks). (**J**) Average roaming activity of top (red) and bottom (blue) 15% of extreme individuals within each of the PC1–3 individuality dimensions in 4-day starved and unstarved wild-type populations. *p < 0.05, ***p < 0.001 (FDR corrected).

The online version of this article includes the following figure supplement(s) for figure 2:

**Figure supplement 1.** Principal component analysis (PCA) and behavioral consistency analyses.

PCA space generated from the shuffled rank dataset (*Figure 2G–J*; *Figure 2—figure supplement 1G, H*), indicating the coexistence of these alternative individuality dimensions. Altogether, these results demonstrate the use of unsupervised analysis for identifying multiple individuality dimensions across development and suggest a broad individuality space within isogenic populations.

## Dopamine buffers behavioral responses to early stress during intermediate developmental stages

Neuromodulatory pathways are known to establish internal behavioral states and modify them based on the environmental context (*Harris-Warrick and Marder, 1991*; *Bargmann, 2012*; *Marder, 2012*; *Kennedy et al., 2014*; *Taghert and Nitabach, 2012*; *Nusbaum and Blitz, 2012*). In particular, the bioamine dopamine was implicated in controlling a wide array of behavioral outputs at various timescales, ranging from minutes and hours, to long-term behavioral patterns that are regulated across life stages (*Marella et al., 2012*; *Omura et al., 2012*; *Sawin et al., 2000*; *Cermak et al., 2020*; *Stern et al., 2017*). In *C. elegans*, dopamine is produced in a specific set of neuronal sites and its effects are known to be mediated by dopamine receptors that are localized to responding neurons (*Sulston et al., 1975*; *Lints and Emmons, 1999*; *Chase et al., 2004*; *Tsalik et al., 2003*).

To ask if dopamine acts across different developmental stages to shape the discontinuous pattern of long-term behavioral responses to early stress and to dissect its temporal requirement, we tracked the behavior of dopamine-deficient *cat-2* animals following exposure to L1 starvation (*Figure 3A*; *Figure 3—figure supplement 1A*). When continuously grown in a food environment, *cat-2* individuals show a long-term roaming activity pattern that is similar to the wild-type population (*Stern et al., 2017*; *Figure 1*; *Figure 3*). However, we found that in contrast to stressed wild-type individuals that show buffering of behavioral responses during the L2, L3, and L4 intermediate stages, *cat-2* individuals that were exposed to early starvation show reduction in average roaming activity across all developmental stages, including during the intermediate stages (*Figure 3B, C*). The behavioral effects of early starvation during mid-development in *cat-2* individuals were not only restricted to animals that were exposed to long starvation periods, as 1 day of early starvation was sufficient to induce a strong reduction in roaming activity during the L2–L4 intermediate stages (*Figure 3D*; *Figure 3—figure supplement 1B–D*). Interestingly, during the L2 intermediate stage the effects on roaming activity patterns were more pronounced during earlier time windows of the stage, suggesting a potential within-stage regulation of behavioral response by dopamine (*Figure 3B, C*; *Figure 3—figure supplement 1C*). However, behavioral effects during L1 and adulthood following early stress were similar in *cat-2* and wild-type (*Figure 3B–D*), implying that dopamine function is mainly required during intermediate developmental stages to buffer alterations in roaming activity in response to a transient early stress.

It was previously shown that during L2 to adulthood, *cat-2* animals have higher instantaneous speed during roaming episodes (*Stern et al., 2017*; *Sawin et al., 2000*; *Figure 3—figure supplement 1E*). We found that unlike stressed wild-type individuals in which roaming speed was decreased mainly during L1 and adulthood (*Figure 1—figure supplement 1G*), *cat-2* mutants show lower speed also across the L2 and L3 stages following stress (*Figure 3—figure supplement 1F*).

To further ask if dopamine supplementation can restore roaming activity following stress, during intermediate developmental stages, we exposed *cat-2* individuals to exogenous dopamine (*Figure 4A–C*; *Figure 4—figure supplement 1A–E*). We found that supplementing dopamine was

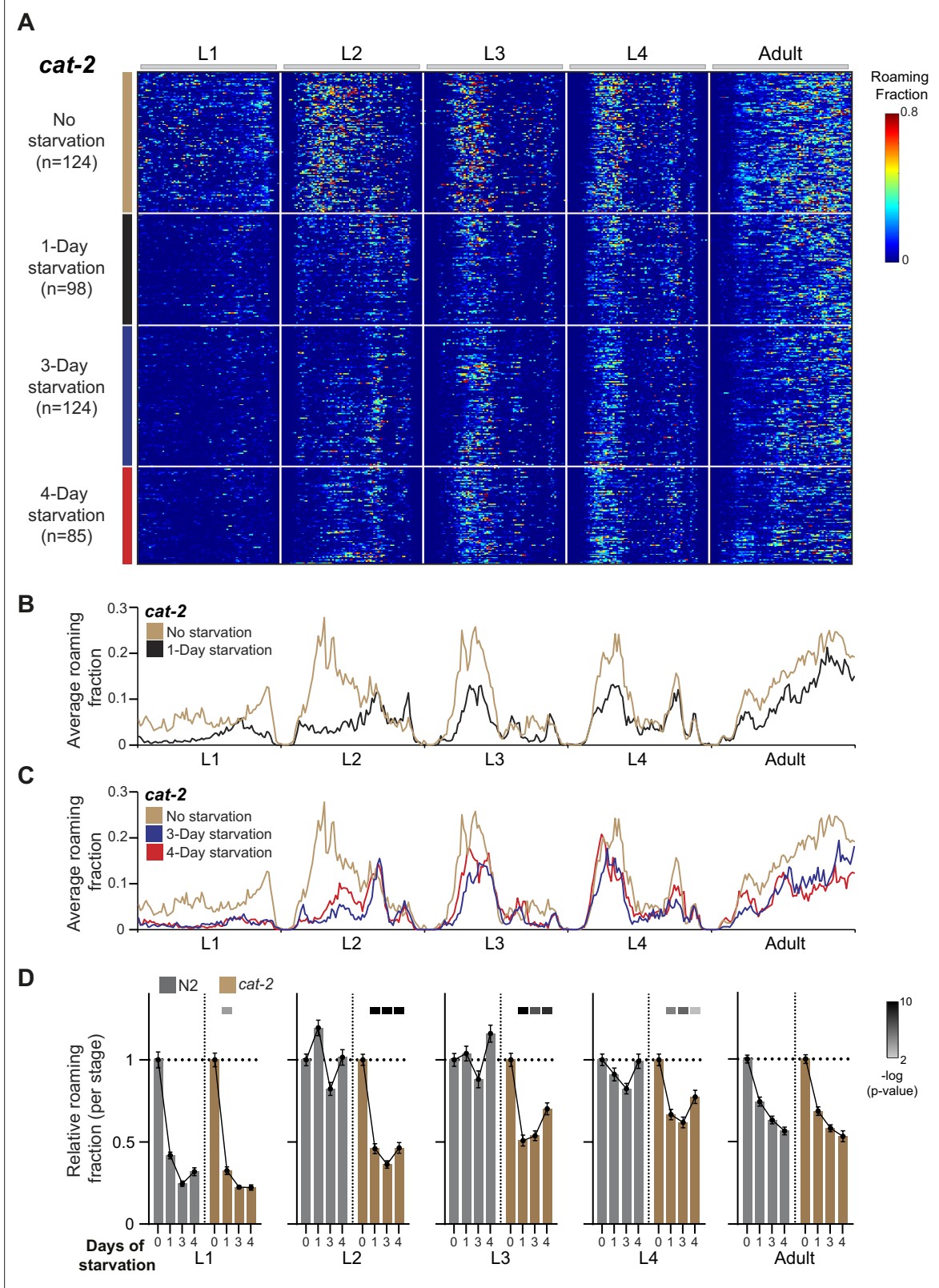

**Figure 3.** Dopamine buffers long-term behavioral effects during intermediate stages of development. (**A**) Roaming and dwelling behavior of *cat-2* animals without early starvation (*n* = 124) and following 1-day (*n* = 98), 3-day (*n* = 124), and 4-day starvation (*n* = 85). Each row indicates the age-normalized behavior of one individual across all developmental stages. The different stages are separated by white lines indicating the middle of the lethargus state. Color bar represents the fraction of time spent roaming in each of the 375 time bins. (**B**) Average roaming fraction of 1-day starved

*Figure 3 continued on next page*

*Figure 3 continued*

*cat-2* animals compared to the unstarved population. (**C**) Average roaming fraction of 3- and 4-day starved *cat-2* animals compared to the unstarved population. (**D**) Average roaming fraction relative to the unstarved population in *cat-2* and wild-type individuals, in each developmental stage. Error bar indicates standard error of the mean. Upper bars indicate statistical significance (Wilcoxon rank-sum test, FDR corrected) of the difference in behavioral effect following early stress between the *cat-2* and wild-type populations (−log(p-value), indicated are p-values <0.01).

The online version of this article includes the following figure supplement(s) for figure 3:

**Figure supplement 1.** Development time and roaming quantification in starved and unstarved *cat-2* individuals.

sufficient to increase only the roaming activity following stress in *cat-2* individuals exposed to 1 and 3 days of starvation, during the L2, L3, and L4 stages (*Figure 4A–C*; *Figure 4—figure supplement 1B, C, F, G*). In contrast, following 1 day of starvation exogenous dopamine did not restore roaming activity during L1 and adulthood (*Figure 4A, C*; *Figure 4—figure supplement 1A, B*) and following 3 days of starvation it only slightly increased roaming in the L1 stage and did not affect roaming activity during adulthood (*Figure 4B, C*; *Figure 4—figure supplement 1A, C*). Overall, these results show that following early and transient starvation, dopamine acts to restrict long-term behavioral alterations in roaming activity, specifically during intermediate developmental windows.

## Specific dopamine receptors function during mid-development to mediate buffering of long-term behavioral responses

The buffering of behavioral effects during the L2, L3, and L4 intermediate developmental stages by dopamine led us to explore the temporal contribution of specific dopamine receptors during these development times. The *C. elegans* dopamine receptor DOP-1 is a D1-like receptor which signal through $G\alpha_{s/olf}$ to activate adenylyl cyclase and DOP-2 and DOP-3 receptors are D2-like receptors which signal via $G\alpha_i$ to suppress adenylyl cyclase (*Chase et al., 2004*; *Sanyal et al., 2004*; *Sugiura et al., 2005*; *Suo et al., 2003*).

To study the independent function of dopamine receptors we analyzed the long-term behavioral effects of early starvation in animals mutant for each of the single dopamine receptors. These analyses showed that each receptor has a different temporal effect on behavioral responses within the intermediate L2–L4 stages (*Figure 4D–I*; *Figure 4—figure supplement 2*). In particular, following 3 days of early starvation, *dop-2* animals showed strong roaming decrease during the L2 and L4 stages, but not during the L3 stage, compared to wild-type (*Figure 4D, G*; *Figure 4—figure supplement 2*). Interestingly, similar to *cat-2* mutants, roaming was mainly decreased in *dop-2* individuals during early time windows within the L2 stage. In addition, *dop-1* individuals showed a roaming decrease during the L2 stage and opposite effects during L3 and L4 (*Figure 4E, H*; *Figure 4—figure supplement 2*) and *dop-3* animals showed weaker overall roaming response during the L2 stage (*Figure 4F, I*; *Figure 4—figure supplement 2*).

Previously, DOP-2 and DOP-3 were shown to function cooperatively (*Cermak et al., 2020*; *Suo et al., 2009*). Therefore, we sought to test if simultaneous alteration of both dopamine receptors will recapitulate the full long-term behavioral effect during intermediate developmental stages, as shown in *cat-2* mutants. We found that following early starvation, *dop-2;dop-3* double mutants showed a decreased roaming activity across all intermediate stages (*Figure 4J, K*; *Figure 4—figure supplement 3*). These results imply that a deficiency in both DOP-2 and DOP-3 receptors is sufficient to recapitulate the behavioral effects in dopamine-deficient individuals during mid-development and suggest that buffering of long-term behavioral responses by dopamine is temporally regulated by the modular function of specific dopamine receptors.

## Serotonin promotes behavioral responses to early stress during early and late developmental stages

To ask if the stage-specific effects of early-life stress on developmental patterns of behavior are an integration of multiple temporal responses that are mediated by different neuromodulators, we also examined serotonin function in shaping long-term behavior following early starvation (*Figure 5A*; *Figure 5—figure supplement 1A*). Under normal growth conditions, serotonin-deficient *tph-1* individuals roam more than wild-type across all developmental stages (*Flavell et al., 2013*; *Stern et al., 2017*; *Figure 5—figure supplement 1B*). We found that contrary to the effects of dopamine on the

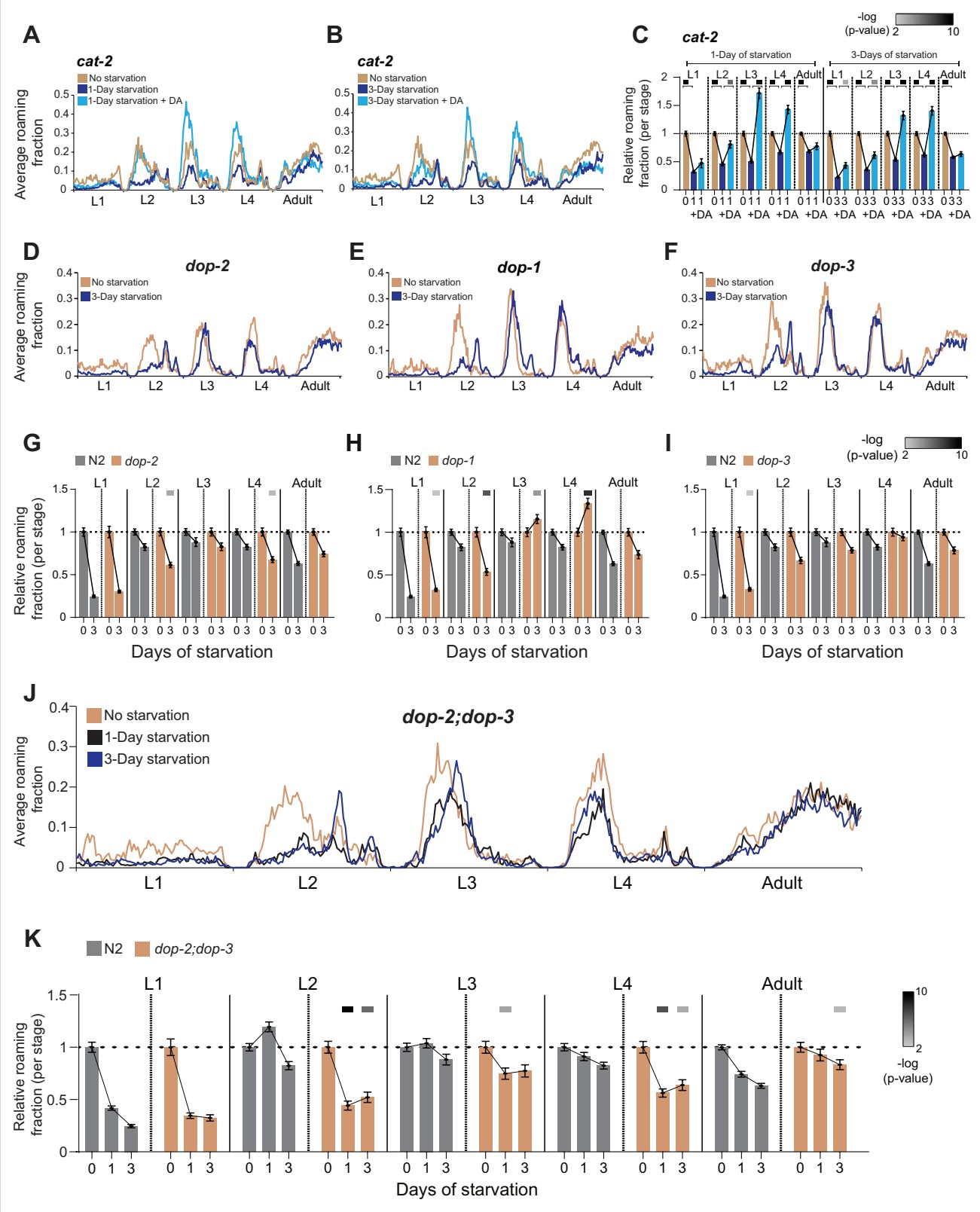

**Figure 4.** Effects of exogenous dopamine and temporally restricted functions of dopamine receptors across intermediate developmental stages. (**A**) Average roaming fraction of unstarved (*n* = 124), 1-day starved (*n* = 98), and 1-day starved with exogenous DA (*n* = 46) *cat-2* populations. (**B**) Average roaming fraction of unstarved (*n* = 124), 3-day starved (*n* = 124), and 3-day starved with exogenous DA (*n* = 50) *cat-2* populations. (**C**) Average roaming fraction relative to the unstarved population in 1- and 3-day starved *cat-2* populations, with or without exogenous DA, in each developmental stage.

*Figure 4 continued on next page*

*Figure 4 continued*

Upper bars indicate statistical significance (−log(p-value), Wilcoxon rank-sum test (FDR corrected), indicated are p-values <0.01. (**D**) Average roaming fraction of 3-day starved (*n* = 145) and unstarved (*n* = 111) *dop-2* populations. (**E**) Average roaming fraction of 3-day starved (*n* = 134) and unstarved (*n* = 73) *dop-1* populations. (**F**) Average roaming fraction of 3-day starved (*n* = 95) and unstarved (*n* = 82) *dop-3* populations. (**G**) Average roaming fraction relative to the unstarved population in *dop-2* and wild-type individuals, in each developmental stage. (**H**) Average roaming fraction relative to the unstarved population in *dop-1* and wild-type individuals, in each developmental stage. (**I**) Average roaming fraction relative to the unstarved population in *dop-3* and wild-type individuals, in each developmental stage. (**J**) Average roaming fraction of 1-day starved (*n* = 62), 3-day starved (*n* = 63), and unstarved (*n* = 69) *dop-2;dop-3* populations. (**K**) Average roaming fraction relative to the unstarved population in *dop-2;dop-3* and wild-type individuals, in each developmental stage. Upper bars in (**G, H, I, K**) indicate statistical significance (Wilcoxon rank-sum test, FDR corrected) of the difference in behavioral effect following early stress between the dopamine receptors mutants and N2 populations (−log(p-value), indicated are p-values <0.01. Error bars indicate standard error of the mean.

The online version of this article includes the following figure supplement(s) for figure 4:

**Figure supplement 1.** Development time and roaming quantification in unstarved, starved, and starved with exogenous DA *cat-2* mutants.

**Figure supplement 2.** Development time and roaming quantification in starved and unstarved dopamine receptors mutants.

**Figure supplement 3.** Development time and roaming quantification in starved and unstarved dopamine receptors double mutants.

buffering of behavioral responses during intermediate stages, *tph-1* individuals that were exposed to 1 day of early starvation maintained their roaming activity during L1 and adulthood (*Figure 5B, D*; *Figure 5—figure supplement 1C*), compared to the strong roaming decrease generated in the wild-type population during these early and late stages. In addition, no significant difference in roaming response was shown during the L2–L4 intermediate developmental stages in the *tph-1* population following 1 day of early starvation (*Figure 5B, D*; *Figure 5—figure supplement 1C*), compared to wild-type.

To test if longer starvation periods early in life will establish behavioral effects during L1 and adulthood we further exposed *tph-1* animals to 3- and 4 days of early starvation. We found that long starvation periods led to a reduction in roaming activity in the L1 stage of *tph-1* animals. However, during the adult stage, *tph-1* individuals were still less responsive to early stress, compared to the strong decrease in roaming in the wild-type (*Figure 5C, D*; *Figure 5—figure supplement 1C, D*). In addition, behavioral responses to early stress were similar in *tph-1* and wild-type individuals during the intermediate L2–L4 stages, indicating that serotonin effects are specific to shaping behavioral responses during L1 and adulthood.

These results show that dopamine and serotonin functions are opposite and segregated across developmental stages in regulating long-term roaming behavior following stress. While dopamine buffers behavioral modifications during intermediate stages of development, serotonin functions to promote behavioral sensitivity to early starvation during the early L1 stage and adulthood. Interestingly, functional segregation among dopamine and serotonin regulation is behavior specific, as the long-term effects of early stress on roaming speed were similar in *tph-1* and *cat-2* individuals (*Figure 5—figure supplement 1E*; *Figure 3—figure supplement 1F*).

## Early-life experience and neuromodulation shape variation in specific individuality dimensions

To ask if early-life experience and neuromodulatory pathways affect specific individuality dimensions to reshape inter-individual variation within populations, we performed the PCA on pooled data across the wild-type, *cat-2*, and *tph-1* populations (*Figure 6—figure supplement 1A–C, E*). We then directly compared inter-individual variation in PC scores within specific individuality dimensions between stressed and unstressed populations of wild-type and neuromodulatory mutant individuals. We found that early starvation modified inter-individual variation in specific individuality dimensions and that wild-type and neuromodulatory mutant populations showed both shared and unique effects on inter-individual variation following the same stressful condition (*Figure 6*; *Figure 6—figure supplement 1F–H*). In particular, both the wild-type and dopamine-deficient *cat-2* populations showed an increase in inter-individual variation in PC scores within the PC3 individuality dimension (double bias switching across development) following starvation, compared to the unstressed population (*Figure 6B, D, F*). In contrast, inter-individual variation in scores within the PC3 dimension was not significantly altered in serotonin-deficient *tph-1* individuals following the same stressful experiences (*Figure 6B, E*). Furthermore, we found that inter-individual variation in scores within the PC6 individuality dimension which

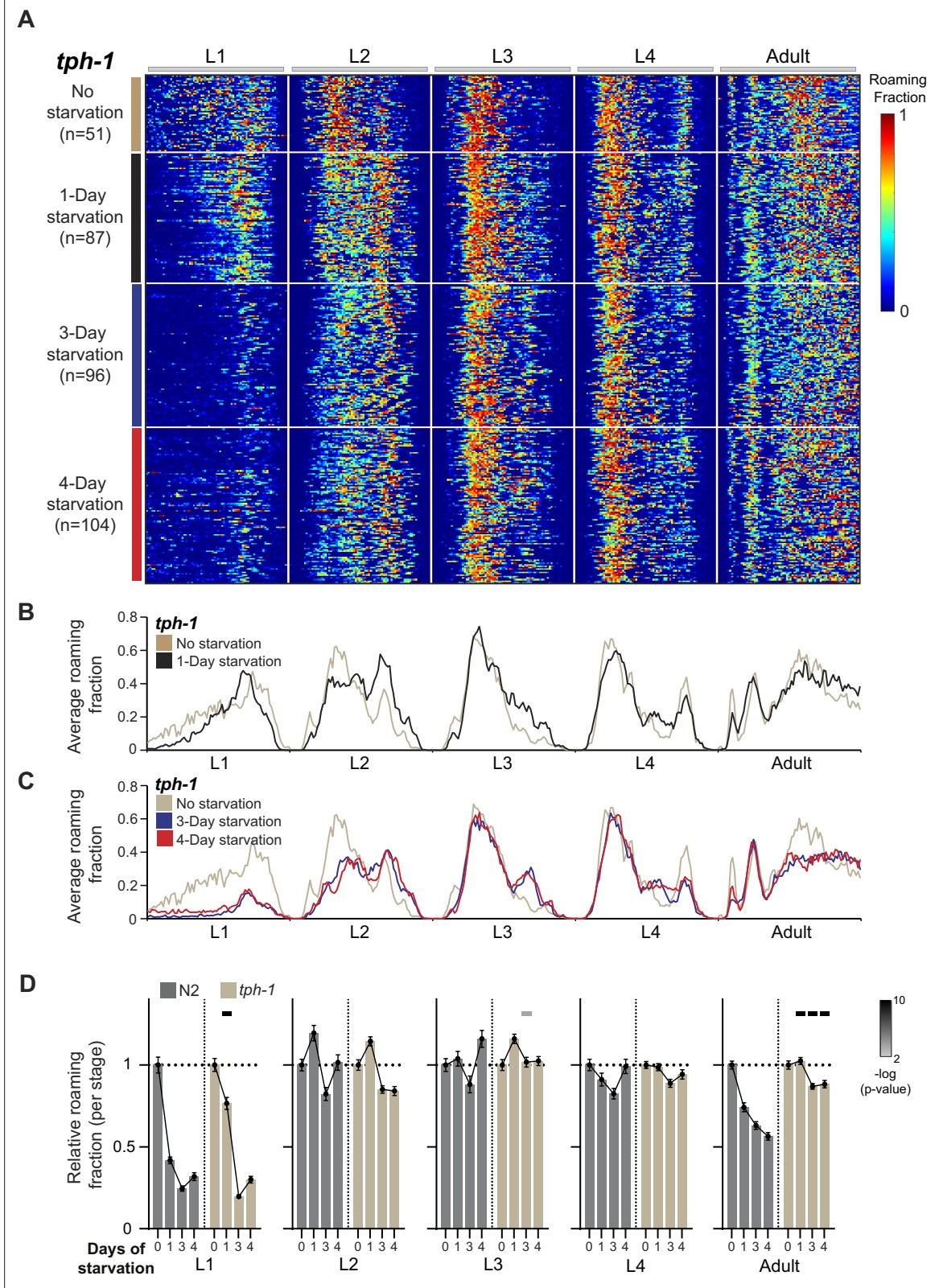

**Figure 5.** Serotonin affects the level of behavioral sensitivity to early stress during early and late developmental stages. (**A**) Roaming and dwelling behavior of *tph-1* animals without early starvation (*n* = 51) and following 1-day (*n* = 87), 3-day (*n* = 96), and 4-day starvation (*n* = 104). Each row indicates the age-normalized behavior of one individual across all developmental stages. The different stages are separated by white lines indicating the middle of the lethargus state. Color bar represents the fraction of time spent roaming in each of the 375 time bins. (**B**) Average roaming fraction of 1-day

*Figure 5 continued on next page*

*Figure 5 continued*

starved *tph-1* animals compared to the unstarved population. (**C**) Average roaming fraction of 3- and 4-day starved *tph-1* animals compared to the unstarved population. (**D**) Average roaming fraction relative to the unstarved population in *tph-1* and wild-type individuals, in each developmental stage. Error bar indicates standard error of the mean. Upper bars indicate statistical significance (Wilcoxon rank-sum test, FDR corrected) of the difference in behavioral effect following early stress between the *tph-1* and N2 populations (−log(p-value), indicated are p-values <0.01).

The online version of this article includes the following figure supplement(s) for figure 5:

**Figure supplement 1.** Development time and roaming quantification in starved and unstarved *tph-1* individuals.

captured multiple bias switching of individuals within developmental stages (L2-Adulthood) was strongly increased following all stressful conditions in the wild-type population, but was decreased after 3 days of early starvation in the *tph-1* population (*Figure 6C–E*). While inter-individual variation in PC6 was affected in opposite directions in the wild-type and *tph-1* populations following stress, *cat-2* individuals did not show altered PC6 inter-individual variation after early-life stress (*Figure 6C, F*). In addition, it was previously shown that unstarved *tph-1* individuals show low levels of behavioral consistency across development compared to the wild-type population, as quantified by the pre-defined consistency index (*Stern et al., 2017*). Similarly, we found that unstarved *tph-1* individuals show lower levels of inter-individual variation in PC scores within the PC1 dimension, which represents individuals with homogenous consistent bias across development, compared to the wild-type population (*Figure 6—figure supplement 1D*; *Figure 6E*). Interestingly, we found that 3- and 4-day starved *tph-1* individuals showed an increase in inter-individual variation in scores within the PC1 individuality dimension, compared to unstressed individuals, while in the wild-type and *cat-2* populations there was no significant change following starvation (*Figure 6A, D–F*). Early starvation affected only a fraction of the identified individuality dimensions as inter-individual variation in scores was not significantly altered following early stress within the PC2, PC4, and PC5 individuality dimensions in all wild-type and neuromodulatory mutant populations (*Figure 6—figure supplement 1F–H*). Overall, these results imply that inter-individual variation in a spectrum of individuality dimensions may be dynamically structured by the early experience of the population and be further modified by its neuromodulatory state.

## Discussion

Spontaneous behavioral patterns across development are structured in time and shaped by the integration of the individual's internal state and its past and current environments. In this work, we studied how developmental patterns of behavior and inter-individual variation are dynamically affected by early-life starvation and the neuromodulatory pathways that organize these long-term behavioral responses. The effects of transient early experiences on neuronal and behavioral states during specific developmental stages were studied across species (*Horn, 1998*; *Kimmel et al., 1974*; *Nakamori et al., 2013*; *Pradhan et al., 2019*; *Remy and Hobert, 2005*; *Wilson and Sullivan, 1994*). However, how transient environmental experiences early in development continuously reshape behavior throughout the full developmental trajectory of the organism is unknown.

Here, we utilized long-term behavioral tracking systems at high spatiotemporal resolution (*Stern et al., 2017*) to analyze and compare long-term alterations of behavioral patterns across and within all developmental stages of *C. elegans*, following transient periods of starvation early in life. As our early starvation paradigm, we let animals hatch into an environment that completely lacks a food source, which leads to developmental arrest during the L1 stage. The L1 arrest state is distinct from the dauer state which is an alternative developmental stage to L3, generated by a combination of environmental stimuli such as overcrowding or limited food availability.

Our results show that early L1 starvation induces stage-specific behavioral responses that are discontinuous across development, manifested by stronger decrease in roaming activity during early and late stages, compared to intermediate developmental stages. These variable influences of early starvation across development time suggest that while the memory of early experiences is maintained to adulthood, behavioral changes are buffered during mid-development.

As imprinting of early memories was shown to have an adaptive value for later stages of life (*Immelmann, 1975*), we hypothesized that neuronal mechanisms actively buffer behavioral alterations at specific development times so as to support the exploratory activity of individuals during critical developmental windows. Building on this idea, we further analyzed the contribution of

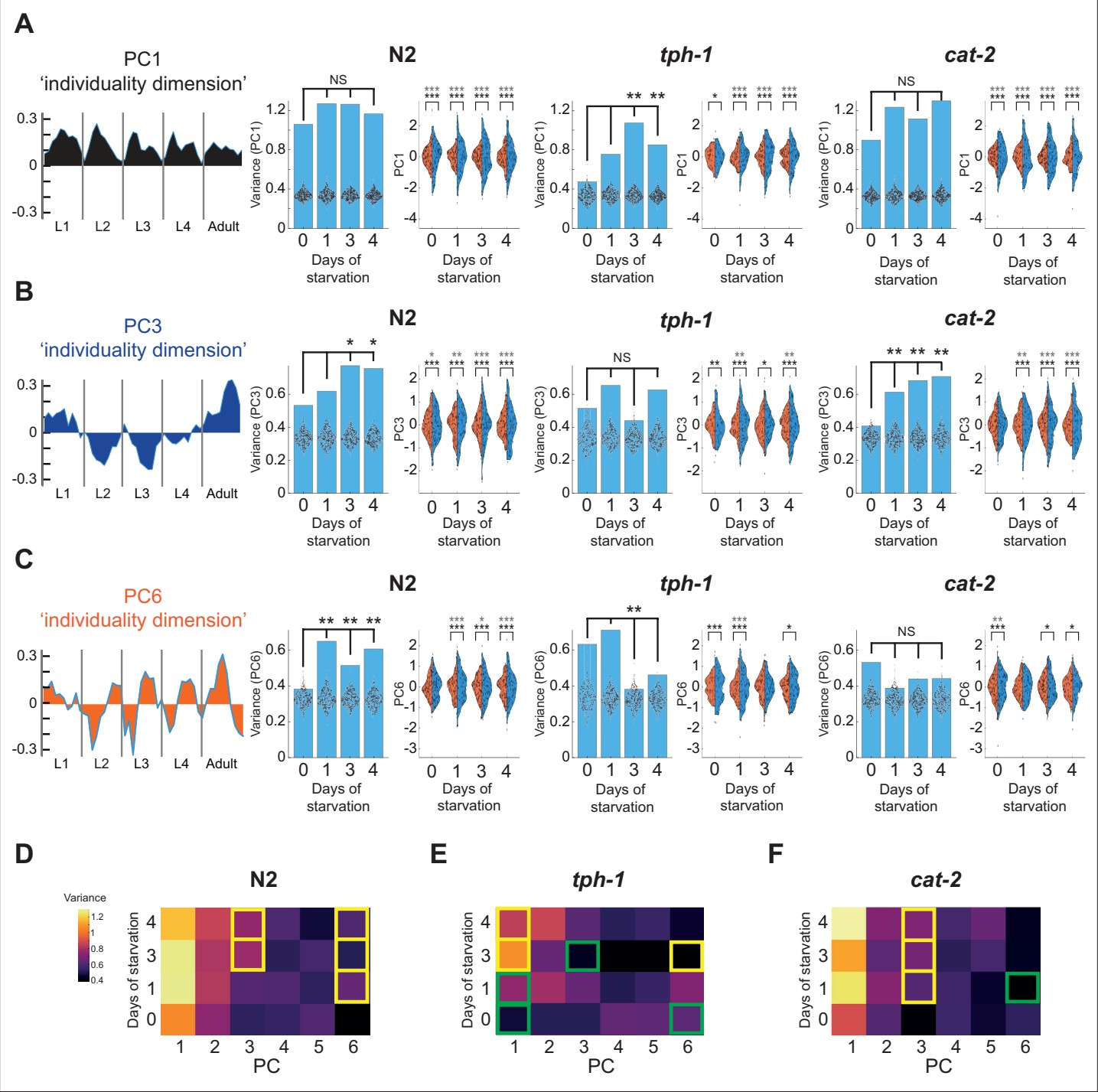

**Figure 6.** Experience-dependent and neuromodulatory effects on variation within individuality dimensions. (**A**) Left: PC1 components in each of the 50 time windows. Bar plots represent inter-individual variation in PC1 individual scores within the wild-type and neuromodulatory mutant populations. p-values in bar plots were calculated using bootstrapping (see Methods) for difference in PC1 variation between starved and unstarved populations. Each dot within bars represents PC1 variation within a shuffled rank dataset in the same principal component analysis (PCA) space (500 repetitions). Distributions show dispersion of PC1 individual scores (blue) within starved and unstarved wild-type and neuromodulatory mutant populations, compared to a shuffled rank dataset in the same PCA space (orange). p-values above distributions were calculated using bootstrapping (see Methods) for significance of difference in PC1 variation to variation of a shuffled dataset in the same PCA space (bottom asterisks) or in a PCA space extracted from the shuffled dataset (upper asterisks). (**B**) Same as (**A**) for PC3. (**C**) Same as (**A**) for PC6. (**D–F**) Heat maps represent inter-individual variation in PC scores within the PC1–6 individuality dimensions in starved and unstarved populations of wild-type (**D**), *tph-1* (**E**), and *cat-2* (**F**) individuals. Significant differences (p < 0.05, FDR corrected) in (**D–F**) are marked for comparisons between starved and unstarved populations of the same genotype (yellow)

*Figure 6 continued on next page*

*Figure 6 continued*

and for comparisons between neuromodulatory mutants and wild-type populations exposed to the same starvation condition (green). *p < 0.05, **p < 0.01, ***p < 0.001 (FDR corrected).

The online version of this article includes the following figure supplement(s) for figure 6:

**Figure supplement 1.** Principal component analysis (PCA) and behavioral consistency analyses in wild-type and neuromodulatory mutants.

neuromodulatory pathways for shaping the stage-specific patterns of behavioral responses across development. Neurotransmitters and hormones were shown to regulate behavioral patterns across development (*Sisk and Foster, 2004*; *Truman, 2005*; *Wigglesworth, 1936*; *Aton et al., 2005*; *Park and Hall, 1998*; *Rehm et al., 2008*). In *C. elegans* populations grown continuously on food, neuromodulators show both consistent and time-dependent behavioral effects at specific developmental windows (*Stern et al., 2017*). We found that following early transient stress, dopamine and serotonin control of long-term behavioral responses are opposite and temporally segregated over development time. Dopamine was required for behavioral buffering during intermediate developmental stages and serotonin established behavioral responses to early stress during early and late developmental stages. In *C. elegans*, dopamine is produced in four pairs of neurons: CEPV, CEPD, ADE, and PDE (*Sulston et al., 1975*; *Lints and Emmons, 1999*) and was shown to be required for controlling locomotory patterns (*Omura et al., 2012*) and coupling of behavioral programs (*Cermak et al., 2020*). In particular, dopamine was shown to decrease the instantaneous speed of worms grown on food, compared to non-food environment (*Sawin et al., 2000*). We found that following early-life starvation, dopamine is required for buffering roaming decrease during intermediate developmental stages. These diverse behavioral effects on different locomotory parameters suggest that dopamine function is variable under different environmental contexts and at different timescales.

By analyzing single dopamine receptors (DOP-1, DOP-2, and DOP-3), we found that their functions are differentially distributed during intermediate developmental stages. We further found that a combination of multiple dopamine receptors is required for establishing behavioral buffering across all intermediate developmental stages (L2–L4). The modular regulation by each of the receptors and their cooperative function imply that dopamine receptors' temporal requirement may be super-imposed in time to mediate buffering of behavioral responses at specific developmental windows. Interestingly, the expression patterns of the three dopamine receptors within the *C. elegans* nervous system are partially overlapping (*Tsalik et al., 2003*; *Chase et al., 2004*; *Sanyal et al., 2004*; *Suo et al., 2003*), raising the possibility that different subnetworks within the nervous system function to temporally regulate behavioral buffering across development. Similarly, the function of serotonin receptors in maintaining patterns of roaming activity in unstressed individuals was also shown to be modular across developmental stages (*Stern et al., 2017*), suggesting a common principle of temporal regulation of behavior by neuromodulatory receptors.

In contrast to dopamine function during intermediate developmental stages, we showed that serotonin promotes behavioral responses to early stress during L1 and adulthood. Under normal growth conditions, serotonin is known to regulate roaming behavior in *C. elegans* across all developmental stages, (*Flavell et al., 2013*; *Stern et al., 2017*) and is required for long- and short-term associative olfactory memory (*Jin et al., 2016*; *Zhang et al., 2005*). In rodents, serotonin and dopamine interact to establish motor patterns (*Sasaki-Adams and Kelley, 2001*). It is plausible that the complexity of long-term behavioral responses to early stress reflects a time integration of the function of multiple neuromodulators, each of them acting at different development times and with different intensity.

Long-term individuality in behavior is observed across species, even within genetically identical populations that were raised in the same environment. However, the effects of early stressful experiences on patterns of individual biases within isogenic populations are less explored. The long-term behavioral tracking of single animals allowed us to ask how early-life stress modifies patterns of inter-individual behavioral variation and whether neuromodulation controls the structure of individuality under stress. Individuality is classically defined as the tendency of an individual to show the same behavioral bias relative to the population over long timescales. We hypothesized that within isogenic populations, individuals may show alternative modes of temporal behavioral biases across development that are not random and represent alternative individuality dimensions.

By using an unbiased approach of dimensionality reduction, we found multiple individuality dimensions that coexist within stressed and unstressed populations. While the main PC1 individuality

dimension recaptured a known individuality dimension of consistent individual biases over time (*Stern et al., 2017*), other PCs identified alternative individuality dimensions that are significant within populations and represent individuals that show switching of behavioral bias, relative to the population, at specific developmental times. These results further extend the view of long-term behavioral individuality, implying a wide spectrum of alternative individual biases within populations (*Tang et al., 2012*; *Werkhoven et al., 2021*). A plausible explanation for the coexistence of multiple individuality dimensions is that, upon stress or another unpredictable environment, it will be beneficial for the population to dynamically reshape the variation across a spectrum of individuality dimensions so as to modify individual strategies and increase the population's chance of survival (*Cooper and Kaplan, 1982*; *Honegger and de Bivort, 2018*).

Neuromodulation was previously shown to affect levels of consistent individual biases (*Honegger et al., 2020*; *Kain et al., 2012*; *Pantoja et al., 2016*; *Stern et al., 2017*). We tested how early-life experiences and neuromodulation shape the identified individuality dimensions across development. Interestingly, we found that inter-individual variation in specific dimensions depends on both the early experience of the population and its neuromodulatory state. An open question is what are the sources of variation within the nervous system that give rise to variation across different individuality dimensions? Underlying differences among individuals may include diversity in gene-expression patterns (*Casanueva et al., 2012*), nervous system structure (*Witvliet et al., 2021*; *Brittin et al., 2021*; *Linneweber et al., 2020*; *Churgin et al., 2021*), and underlying persistent differences in neuromodulatory parameters that have phenotypic effects under specific conditions (*Marder et al., 2022*). It is plausible that some of this variation, which is partially stochastic by nature, may generate different behavioral biases of individuals within isogenic populations. While we quantify proportional behavioral effects in neuromodulatory mutants following starvation, relative to the baseline levels in the unstarved population, these behavioral effects may potentially reflect a more complicated interaction between neuromodulation and stress, altering baseline levels and deviations from baseline. More generally, taking into consideration the complex relationship between different effects within a non-linear system (*Félix and Barkoulas, 2015*), part of the temporal- or inter-individual variation in behavior may not directly rely on differences in neuromodulatory states over time or across individuals, but rather on the modification of behavioral sensitivity to underlying variations by specific neuromodulatory and environmental perturbations (*Maloney, 2021*).

The behavioral patterns explored in this study represent only a small fraction of the behavioral space available to the organism (*Ahamed et al., 2021*; *Anderson and Perona, 2014*; *Brown and de Bivort, 2018*; *Schwarz et al., 2015*). In addition, while we were able to extract behavioral changes in specific roaming parameters across all developmental stages, the lower spatial imaging resolution during the L1 stage may limit our ability to detect smaller modifications in behavior during this stage. We anticipate that an extended supervised and unsupervised behavioral classification across development will shed light on the overall reorganization of individuality dimensions and the contribution of both internal neuronal states and external environments to the diversity in long-term behavioral structures within populations.

## Materials and methods
### Growth conditions and starvation protocol

*C. elegans* worms were maintained on NGM agar plates, supplemented with *E. coli* OP50 bacteria as a food source. For behavioral tracking, we imaged single individuals grown in custom-made laser-cut multi-well plates. Each well (10 mm diameter) was seeded with a specified amount of OP50 bacteria (10 µL of 1.5 OD) that was UV-killed before the experiment to prevent bacterial growth. For the starvation experiments, eggs were collected from isogenic populations using a standard bleaching protocol, into an agar plate without OP50 bacteria. Newly hatched L1 larvae were starved for a specified time window (L1 arrest of 1, 3, or 4 days) before being transferred to the imaging multi-well plates. For tracking behavior without early L1 starvation, animals were monitored immediately after hatching in the multi-well plates. For DA supplementation experiments, animals were exposed during starvation to 15 mM DA for 1 day before transferring them to OP50 bacterial food supplemented with 600 mM DA.

## *C. elegans* strains

Strains used in this study:

Wild-type Bristol N2
MT15434 *tph-1* (mg280) II
CB1112 *cat-2* (e1112) II
LX645 *dop-1* (vs100) X
LX702 *dop-2* (vs105) V
LX703 *dop-3* (vs106) X
LX704 *dop-2* (vs105) V; *dop-3* (vs106) X

## Imaging system

Longitudinal behavioral imaging was performed using custom-made imaging systems. Each imaging system consists of an array of six 12 MP USB3 cameras (Pointgrey, Flea3) and 35-mm high-resolution objectives (Edmund optics) mounted on optical construction rails (Thorlabs). Each camera images up to six wells, each containing an individual grown in isolation. Movies are captured at 3 fps with a spatial resolution of ~9.5 μm. For uniform illumination of the imaging plates we used identical LED backlights (Metaphase Technologies) and polarization sheets. To tightly control the environmental parameters during the experiment, imaging was conducted inside a custom-made environmental chamber in which temperature was controlled using a Peltier element (TE technologies, temperature fluctuations in the range of 22.5 ± 0.1°C). Humidity was held in the range of 50 ± 5% with a sterile water reservoir and outside illumination was blocked, keeping the internal LED backlights as the only illumination source. Movies from the cameras were captured using commercial software (FlyCapture, Pointgrey) and saved on two computers (3 cameras per computer; each computer has at least 8-core Intel i7/i9 processor and 64 GB RAM).

## Imaging data processing for extracting locomotion trajectory

To extract behavioral trajectories of animals across the experiment, captured movies were analyzed by custom-made script programmed in MATLAB (Mathworks, version 2019b) (*Stern et al., 2017*). In each frame of the movie and for each behavioral arena, the worm is automatically detected as a moving object by background subtraction, and its XY position is logged (center of mass). In each experiment, 600,000–1,000,000 frames per individual are analyzed using ~50 processor cores in parallel to reconstruct the full behavioral trajectory of individuals over days of measurements across development. The total time of image processing was 3–7 days per experiment. Egg hatching time of each individual in the experiment is automatically marked by the time when activity can be detected in the behavioral arena. The middle of the lethargus periods, in which animals stop their locomotion and molt, were defined as the transition points between different stages of development (based on 10 s timescale speed trajectories over time, smoothed over 300 frames). To synchronize temporal behavioral trajectories of different individuals we age-normalized individuals by dividing the behavioral trajectory of each life stage into a fixed number of time windows.

## Behavioral parameters quantification

For each individual, we differentiate between roaming and dwelling states by averaging speed (μm/s) and angular velocity (absolute deg/s) over 10 s using a rolling time window, and generating a 2D probability distribution of these two behavioral parameters for all intervals in each time bin along the experiment (50 × 50 bins distribution, speed bin size: 7.59 μm/s, angular velocity bin size: 3.6 deg/s) (*Stern et al., 2017*). Drawing a diagonal through the probability distribution separated roaming and dwelling states, such that intervals in the distribution bins below the diagonal were classified as roaming intervals and intervals in bins above the diagonal were classified as dwelling intervals (*Ben Arous et al., 2009*; *Flavell et al., 2013*; *Stern et al., 2017*). The behavior of each animal over time could be quantified as a sequence of roaming and dwelling intervals. The fraction of time spent roaming of the individual in a time bin represents the fraction of these intervals classified as roaming states within a given time bin. For each developmental stage, we examined the two-dimensional probability distribution of the whole population and changed the slope of the diagonal to classify roaming and dwelling appropriately (slopes: 5, 2.5, 2.3, 2, and 1.5 for the L1, L2, L3, L4, and adult

stages, respectively). Based on these roaming and dwelling classifications we further quantified the average instantaneous speed of the animal during roaming episodes (μm/s).

## Unsupervised quantification of temporal individuality dimensions

### Ranking and behavioral bias

Individuals within the population were ranked based on their roaming behavior in 50 time bins (10 per stage), relative to the population measured within the same experiment. More explicitly, within each experiment, individuals were ranked in each time bin by the fraction of time within the bin spent roaming. Ties were resolved as fractional ranks (1 2.5 2.5 4 ranking). This produces a rank $r_{i,k}$ for the $i$th individual in the $k$th time bin, between 1 and $n_i$, where $n_i$ is the number of individuals measured in the experiment which includes individual $i$. These ranks were normalized to obtain *bias* values between $-1$ and 1, as $b_{i,k} = \frac{2}{n_i}\left(r_{i,k} - \frac{1}{2}\right) - 1$. Thus, a bias $b_{i,k} = 0$ is obtained when the roaming fraction of worm $i$ in bin $k$ is the median roaming fraction for that experiment. A positive bias occurs in a time bin where a worm roams more than the median roaming fraction for that time bin across its experiment, and a negative bias where it roams less than the median. Particularly, in each time bin, the worms with the highest and lowest roaming fraction within an experiment have biases $(1 - 1/n)$ and $(-1 + 1/n)$, respectively, where $n$ is the number of worms in the experiment.

### Identification of temporal bias patterns

To identify temporal individual biases that are dominant within the isogenic populations, we performed PCA on individuals' biases across time bins.

This analysis represents each individual's sequence of biases $\boldsymbol{b}_i = (b_{i,k})_{k=1}^{50}$, as a weighted sum of principal components (PCs),

$$\boldsymbol{b}_i = \Sigma_{k=1}^{50} t_{i,k}\boldsymbol{w}_k; \quad t_{i,k} = \boldsymbol{b}_i \cdot \boldsymbol{w}_k \qquad (1)$$

where $\boldsymbol{w}_k$ is the $k$th PC, $t_{i,k}$ the $k$th PC score for the $i$th individual. Note that *equation (1)* does not include a mean term, since the mean of all biases at each time bin is zero by construction. The first PC $\boldsymbol{w}_1$ is the direction where the variance of the population is highest (namely, the unit vector for which the variance of the dot product $\boldsymbol{b}_i \cdot \boldsymbol{w}_1$ across the population is maximized). The second PC $\boldsymbol{w}_2$ is the direction of highest variance in the subspace orthogonal to $\boldsymbol{w}_1$, and so on. The PCs are obtained as eigenvectors of the covariance matrix of the input vectors $\boldsymbol{b}_i$.

When computing PCA across several experimental conditions, each individual bias vector $\boldsymbol{b}_i$ was weighted in inverse proportion to the number of individuals in the same condition (strain and starvation level), so that each condition has equal weight. Specifically, the variance which is maximized by the PCs is the weighted variance of the bias vectors, where $\boldsymbol{b}_i$ is assigned the weight $1/n_i$. In practice, this is achieved by computing the PCs as eigenvectors of the *weighted* covariance matrix.

Early principal components thus represent the temporal patterns of individual biases $\boldsymbol{b}_i$ which account for the most variance in the rank sequences.

Statistical significance of PCs was assessed by comparing the variances of PC scores generated from the real individual rank dataset to variances calculated from a randomly shuffled rank dataset, where ranks in each time bin were shuffled independently (500 repetitions). This test identifies which PCs account for a higher fraction of variance than expected by chance. Inter-individual variance in each PC score was calculated as a dispersal parameter of the population for each PC individuality dimension. These score variances were compared to the same-numbered PC in each shuffled dataset in two ways: (1) by performing PCA on the shuffled dataset and computing PC scores of the shuffled dataset in its PCA space and (2) by computing PC scores of the shuffled dataset in the PCA space of the original dataset.

To quantify significant differences in PC score inter-individual variance between conditions, we used a permutation test where individuals in each pair of conditions were randomly reassigned to two populations of the same size. Significance values were computed from 1000 such reassignments for each pair of conditions. The test was repeated multiple times to verify the robustness of the analysis.

### Quantification of individual consistency index

Individuals within the population were ranked based on their behavior in 50 time bins (10 per stage). We then quantified the homogeneous consistent bias in the individual's behavior relative to the population

(*Stern et al., 2017*) by calculating for each individual the $\log_2$(number of time bins in which the individual's roaming fraction is higher than the population median/number of time bins in which the individual's roaming fraction is lower than the population median) (consistency index). This measure gives positive values to individuals that tend to have positive bias (higher than median roaming fraction) across time, negative values to individuals that tend to have negative bias across time, and values close to 0 for individuals that do not show any bias toward higher or lower roaming fraction. Inter-individual variance in consistency index was calculated as a dispersal parameter of the population which indicates the overall consistent behavioral bias in the population. To quantify significant differences in inter-individual variance of behavioral consistency between conditions, we used a permutation test, as used for comparing variance of PC scores (see 'Identification of temporal bias patterns').

## Quantification of worm size

To measure the worm's size in each frame, a cropped image of size 151 by 151 pixels around the detected center of mass was used. First, background subtraction was performed in each frame, using the same method as in *Stern et al., 2017*: The typical background was estimated in each input video (approx. 11.5 min) by averaging 8 equally spaced sample frames. Each frame $f$ in the $i$th video, with grayscale levels represented as 8-bit values in the range 0–255, was normalized as $\widetilde{f} = (f - b_{i+8} + 100)/256$, where $b_{i+8}$ is the background estimate for the $(i + 8)$th video. The worm's contour was then found in each background-subtracted frame using a fixed grayscale threshold of 0.34, and the number of pixels enclosed by the contour was computed.

For a more robust estimate, a running median was applied with a window length of 10 min (301 frames). Specifically, the area estimate in the $i$th frame was obtained as the median of raw pixel counts in frames $i - 150$ to $i + 150$, disregarding any frames within this range where worm detection had failed or where contour computation produced no closed contour or multiple closed contours. This step helps smooth out temporary changes to the worm's apparent size due to imaging noise or changes in posture, as well as errors due to the worm being partially outside the imaging area. For comparing size-matched individuals we quantified statistical difference in roaming parameters within 20 running size windows (width: 10% of range).

## Acknowledgements

We thank Cori Bargmann, Sagi Levy, and the members of our laboratory for comments on the manuscript. Some strains were provided by the CGC, which is funded by the NIH Office of Research Infrastructure Programs (P40 OD010440). This project has received funding from the European Research Council (ERC) under the European Union's Horizon 2020 research and innovation programme (grant agreement No 851634).

---

## Additional information

### Funding

| Funder | Grant reference number | Author |
| --- | --- | --- |
| HORIZON EUROPE European Research Council | ERC-STG-2019 | Shay Stern Yuval Harel Reemy Ali Nasser |

The funders had no role in study design, data collection, and interpretation, or the decision to submit the work for publication.

### Author contributions

Reemy Ali Nasser, Conceptualization, Data curation, Formal analysis, Investigation, Methodology, Writing - original draft, Writing - review and editing; Yuval Harel, Conceptualization, Data curation, Software, Formal analysis, Investigation, Methodology, Writing - original draft, Writing - review and editing; Shay Stern, Conceptualization, Supervision, Funding acquisition, Investigation, Methodology, Writing - original draft, Writing - review and editing

## Author ORCIDs

Shay Stern ⬤ http://orcid.org/0000-0002-9576-7938

## Decision letter and Author response

Decision letter https://doi.org/10.7554/eLife.84312.sa1
Author response https://doi.org/10.7554/eLife.84312.sa2

## Additional files

### Supplementary files

• MDAR checklist

### Data availability

Behavioral datasets have been deposited in Mendeley at https://doi.org/10.17632/fgsyppvpnc.1 and https://doi.org/10.17632/kxrcmtyfr6.1. Code of individuality analysis was deposited in https://github.com/yha/ElegansIndividuality (copy archived at *Harel, 2023*).

The following datasets were generated:

| Author(s) | Year | Dataset title | Dataset URL | Database and Identifier |
|---|---|---|---|---|
| Ali Nasser R, Harel Y, Stern S | 2023 | Behavior_Early_Stress_2 | https://doi.org/10.17632/fgsyppvpnc.1 | Mendeley Data, 10.17632/fgsyppvpnc.1 |
| Ali Nasser R, Harel Y, Stern S | 2023 | Behavior_Early_Stress | https://doi.org/10.17632/kxrcmtyfr6.1 | Mendeley Data, 10.17632/kxrcmtyfr6.1 |

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
