## [Editor Report]

Early life stress can have profound effects on animal behavior, including potential influences on individuality. Here, the authors use a rich new dataset to convincingly demonstrate that the behavioral consequences of early life stress in *C. elegans* can be buffered by neuromodulators previously implicated in patterns of individuality. While much remains to be learned about the mechanisms by which stress might influence individuality, these studies report important advances that will be of interest to neurobiologists studying interactions between behavior, neuromodulation, stress, and individuality.

---

## [Decision Letter]

**Decision letter after peer review:**

Thank you for submitting your article "Early-life experience reorganizes neuromodulatory regulation of stage-specific behavioral responses and individuality types during development" for consideration by *eLife*. Your article has been reviewed by 3 peer reviewers, one of whom is a member of our Board of Reviewing Editors, and the evaluation has been overseen by Piali Sengupta as the Senior Editor. The following individual involved in the review of your submission has agreed to reveal their identity: Ryan T Maloney (Reviewer #2).

Essential revisions:

All three reviewers find your paper to be interesting and a potentially valuable contribution to the field. However, there are a number of concerns that would need to be addressed in a revision. None of these require new experiments. In addition to these essential revisions, each reviewer provides additional detailed feedback below that may be useful in your revision.

1. As reviewer 2 notes (point 1), some of the claims you make about changes in the temporal structure of within-stage behavior have not been rigorously tested. Please carry out additional analyses (or temper your conclusions) to address this point.

2. Reviewer 2 raises a concern (point 2) about the interpretation of PC1 inter-individual variance data in tph-1 mutants (Figure 6D). Please provide a more rigorous test of the proposal that PC1 variance increases with starvation in tph-1 mutants.

3. In the Discussion, please address the issues raised by reviewer 2 about potential confounds associated with baseline effects and non-linear interactions (point 3).

4. Please address the technical concerns of reviewer 3's point 3, either with additional analyses or edits to the text.

5. While your title and abstract emphasize relationships between early-life stress and individuality, the data don't provide strong support for this interaction. Please consider reviewer 3's point 4, which could be addressed with new analyses and/or changes to the text.

*Reviewer #1 (Recommendations for the authors):*

I find this paper to be interesting and solid, but the level of insight that emerges is somewhat limited.

*Reviewer #2 (Recommendations for the authors):*

This was a well written paper on an interesting and well designed project. I hope these comments prove constructive to increasing the clarity and rigor of the paper's arguments without requiring undue delay to final publication.

1) Lack of quantitative analysis for effects within developmental stages.

This should be addressed, presumably with some combination of confidence intervals on the plots and quantitative tests for subdivided developmental stages as necessary to support any claims in the test relating to differences below the level of average roaming fraction per stage.

Specifically:

(1C-D, pg 4 " while 1 day of early starvation modified the temporal structure of activity peaks within the L2 stage") and in describing the apparent differences within a phase for L2-L4 in cat-2 animals (Figure 3B-C).

Ideally the comparison would be done in such a way to prevent any potential artifacts that might arise from normalization (or showing that they persist with or without normalization, or an alternative method of normalization) if possible.

2) Incorrect inferences from differences in significance.

The key problematic claim is:

"However, we found that following long starvation periods (3 and 4 days), inter-individual variation in PC1 type was not significantly different in tph-1 individuals, compared to the wild-type population (Figure 6D). The increase in PC1 inter-individual variation in tph-1 individuals following stress indicates that early starvation experiences may generate extreme behavioral consistency in a specific neuromodulatory context where consistency levels are initially low."

These should be addressed properly before publication-at minimum by testing for significance between the differences between starvation states across conditions (though a linear model looking at Starvation x Genotype effects would work as well). A smaller related change is the caption for figure 5 states that "Serotonin is required for behavioral response during early and late developmental stages"-as shown clearly in figure s7b, serotonin deficient animals do show a behavioral response (albeit attenuated) to starvation.

If the expected difference doesn't emerge from proper statistical analysis, claims (e.g. "further identified experience-dependent effects on their composition.") should be revised accordingly

*Reviewer #3 (Recommendations for the authors):*

Strength:

1) This study provides a very detailed analysis of a single behavioral parameter across the entirety of development from hatch to adulthood, and provides the foundation for many future interesting questions to be asked.

Weakness:

1) Much of the study is mainly descriptive, and the authors perform surface-level examination of the underlying mechanisms of how early-life starvation regulates developmental behavioral trajectory by examining neurotransmitter biosynthesis null mutants. While this provides a good basis to address how early life experience may shape development, the authors stopped shortly of deeper mechanistic investigations such as the critical tissue/cell types of dopamine/serotonin's effects as well as the temporal windows of their necessity/sufficiency. For example, the authors can conduct cell-specific rescue/depletion experiments of cat-2/dop-2/dop-3 to address the spatial requirement/necessary circuit for dopamine, while supplementation or AID (auxin-induced degron) experiments can address the temporal requirement of dopamine/serotonin. These results will provide deeper mechanistic insights and a better understanding of how these neurotransmitter systems contribute to the effects of early-life starvation on roaming behavior.

2) The study largely focuses on a single aspect of behavior: roaming vs. dwelling. How early starvation affects this one behavior parameter and how dopamine/serotonin play a role in it may not be broadly applicable to other behavioral parameters. This was even demonstrated in the manuscript itself, as the authors also looked another related behavioral parameter, speed during roaming, and found different effects of early-life starvation and of dopamine/serotonin on this parameter compared to their effects of average roaming fraction (most prominently demonstrated in Figures S4D vs. C and S7C vs B, but also in Figures S1E vs D and S3C vs. B). The authors are using a very unique technique to be able to capture the entirety of the animal's development, and there are likely many other parameters that can be extracted from this valuable resource. The examination of many different parameters will lead to a better overall picture of how behavior is influenced by early life experiences.

3) There are some confounding factors/technical limitations that are not considered or clearly presented in the manuscript that may alter overall interpretation of the data/conclusions.

a. It is unclear from the current presentation of data that the authors have sufficient resolution to accurately calculate the behavioral parameters of younger, smaller worms, particularly those at the L1 stage. This was evident in the authors' own data that cat-2 mutation did not increase the speed of the L1 worms but at all other stages, and could be as a result of decreased sensitivity/resolution at this stage. The authors should provide evidence that their analysis provides sufficient resolution to measure behavioral features of L1 animals by comparing their analysis/results to more high resolution approaches to validate some of their results. For example, control and cat-2 mutant L1 in their approach vs. a higher resolution approach.

b. Somewhat related to a, the animals change dramatically in size from L1 to the adult stage. This was not taken into account of the calculation of roaming vs. dwelling behavior. Does altered size affect roaming vs dwelling behavior? If so, does starvation and dopamine/serotonin affect size of the animals at different stages, and does this fully or partially explain their effects on roaming/dwelling behavior? The authors should be able to extract length/width data from their recordings and address these questions. This would significantly affect the interpretation of the current manuscript.

4) By the title and abstract of the manuscript, the authors promises to answer the question of how early life starvation affects individuality. However, there is no clear presentation/conclusion in the text/figures of how early life stress affect individuality. For example, are there more/less variation across different principal components after early-life starvation as shown in Figure 2/S2? Rather the authors focused on how dopamine/serotonin affect individually, which was already previous reported in a previous manuscript (Stern et al., 2017). The authors already have many of these analyses presented in the current manuscript (Figure 2, 6, S2, S8), but need to directly compare whether starvation affects variation/individuality in roaming behavior.

---

## [Author Response]

Essential revisions:All three reviewers find your paper to be interesting and a potentially valuable contribution to the field. However, there are a number of concerns that would need to be addressed in a revision. None of these require new experiments. In addition to these essential revisions, each reviewer provides additional detailed feedback below that may be useful in your revision.1. As reviewer 2 notes (point 1), some of the claims you make about changes in the temporal structure of within-stage behavior have not been rigorously tested. Please carry out additional analyses (or temper your conclusions) to address this point.

As the reviewer suggested, in the revised manuscript we include additional analyses of roaming fraction differences across shorter time-windows, demonstrating within-stage changes in temporal behavioral structures (Figure 1 —figure supplement 1E; Figure 3 —figure supplement 1C). In addition, as described below, we temper and rewrite our conclusions to describe these results more clearly (now- “…while 1 day of early starvation modified within-stage temporal behavioral structures by shifting roaming activity peaks to later time-windows during the L2 and L3 stages…” in p. 4 and “Interestingly, during the L2 intermediate stage the effects on roaming activity patterns were more pronounced during earlier time-windows of the stage…” in p. 8).

2. Reviewer 2 raises a concern (point 2) about the interpretation of PC1 inter-individual variance data in tph-1 mutants (Figure 6D). Please provide a more rigorous test of the proposal that PC1 variance increases with starvation in tph-1 mutants.

In the revised manuscript we provide a direct comparison of PCs inter-individual variances between starved and unstarved populations (Figure 6; Figure 6 —figure supplement 1). These analyses directly demonstrate changes in inter-individual variation in specific PC dimensions following starvation, including the increase in PC1 inter-individual variation in *tph-1* mutants following 3- and 4-days of starvation (Figure 6A,E).

3. In the Discussion, please address the issues raised by reviewer 2 about potential confounds associated with baseline effects and non-linear interactions (point 3).

In the discussion part of the revised manuscript we address the issues of mixed effects of neuromodulation and stress on baseline levels and deviations from baseline, as well as putting our results in the context of non-linear systems, in which behavioral sensitivity to underlying variations may be modified by specific neuromodulatory and environmental perturbations (Discussion, p. 16).

4. Please address the technical concerns of reviewer 3's point 3, either with additional analyses or edits to the text.

In the revised manuscript we include additional analyses to control for size differences (based on new individual size extraction), showing behavioral modifications across different conditions/genotypes also in size-matched individuals (within the same size range) (Figure 1 —figure supplement 1F; Figure 3 —figure supplement 1D,E; Figure 5 —figure supplement 1B,D). We also made edits to the text to describe these results (Methods p. 21 and Results section). In addition, as described below, while we capture images with sufficient spatial resolution to demonstrate roaming effects in small L1 larvae (roaming fraction and roaming speed, results from this paper and Stern et al. 2017), we agree with the reviewer that other milder behavioral modifications may be harder to capture because of the relatively lower spatial resolution of these young animals. We now indicate this point of limited spatial resolution during L1 in the text (Discussion p.16).

5. While your title and abstract emphasize relationships between early-life stress and individuality, the data don't provide strong support for this interaction. Please consider reviewer 3's point 4, which could be addressed with new analyses and/or changes to the text.

To address this point, in the revised manuscript we now include a systematic and direct comparison of PCs inter-individual variation between stressed and unstressed populations (within wild-type and neuromodulatory mutants) demonstrating significant changes in variation in specific PC individuality dimensions following early life stress (Figure 6; Figure 6 —figure supplement 1). We further made edits to the text to describe these effects of early-life stress on individuality dimensions (Results p. 11 and abstract).

In addition to the points above, we included additional data of DA supplementation experiments of 1-day starved individuals and updated the experiments data of DA supplementation to 3-day starved individuals to maintain a more robust and comparable DA supplementation protocol across both conditions (Figure 4A-C).

Reviewer #2 (Recommendations for the authors):This was a well written paper on an interesting and well designed project. I hope these comments prove constructive to increasing the clarity and rigor of the paper's arguments without requiring undue delay to final publication.1) Lack of quantitative analysis for effects within developmental stages.This should be addressed, presumably with some combination of confidence intervals on the plots and quantitative tests for subdivided developmental stages as necessary to support any claims in the test relating to differences below the level of average roaming fraction per stage.Specifically:(1C-D, pg 4 " while 1 day of early starvation modified the temporal structure of activity peaks within the L2 stage") and in describing the apparent differences within a phase for L2-L4 in cat-2 animals (Figure 3B-C).Ideally the comparison would be done in such a way to prevent any potential artifacts that might arise from normalization (or showing that they persist with or without normalization, or an alternative method of normalization) if possible.

As the reviewer suggested, we added additional analyses of behavioral effects across shorter time windows, demonstrating within-stage effects on behavioral structure, below the level of average roaming activity per stage (Figure 1 —figure supplement 1E; Figure 3 —figure supplement 1C). In addition, we temper and rewrite our conclusions to specifically describe these effects (now- “…while 1 day of early starvation modified within-stage temporal behavioral structures by shifting roaming activity peaks to later time-windows during the L2 and L3 stages…” in p. 4 and “Interestingly, during the L2 intermediate stage the effects on roaming activity patterns were more pronounced during earlier time-windows of the stage…” in p. 8).

2) Incorrect inferences from differences in significance.The key problematic claim is:"However, we found that following long starvation periods (3 and 4 days), inter-individual variation in PC1 type was not significantly different in tph-1 individuals, compared to the wild-type population (Figure 6D). The increase in PC1 inter-individual variation in tph-1 individuals following stress indicates that early starvation experiences may generate extreme behavioral consistency in a specific neuromodulatory context where consistency levels are initially low."These should be addressed properly before publication-at minimum by testing for significance between the differences between starvation states across conditions (though a linear model looking at Starvation x Genotype effects would work as well). A smaller related change is the caption for figure 5 states that "Serotonin is required for behavioral response during early and late developmental stages"-as shown clearly in figure s7b, serotonin deficient animals do show a behavioral response (albeit attenuated) to starvation.If the expected difference doesn't emerge from proper statistical analysis, claims (e.g. "further identified experience-dependent effects on their composition.") should be revised accordingly

In the revised manuscript we include a direct test of changes in inter-individual variation in specific PC individuality dimensions between starved and unstarved individuals, within the wild-type and mutant populations. These comparisons show significant effects of early starvation on inter-individual variation in specific PC dimensions (Figure 6 and Figure 6 —figure supplement 1), including the increase in variation in PC1 dimension following 3 and 4 days of early starvation in *tph-1* mutants. In addition, we made edits to the text (main text and abstract), based on these new analyses, to better describe these effects of early stress on variation. In addition, as the reviewer suggested, we changed the caption of Figure 5 (now – “Serotonin affects the level of behavioral sensitivity to early stress during early and late developmental stages”) to describe this result more accurately.

Reviewer #3 (Recommendations for the authors):Strength:1) This study provides a very detailed analysis of a single behavioral parameter across the entirety of development from hatch to adulthood, and provides the foundation for many future interesting questions to be asked.Weakness:1) Much of the study is mainly descriptive, and the authors perform surface-level examination of the underlying mechanisms of how early-life starvation regulates developmental behavioral trajectory by examining neurotransmitter biosynthesis null mutants. While this provides a good basis to address how early life experience may shape development, the authors stopped shortly of deeper mechanistic investigations such as the critical tissue/cell types of dopamine/serotonin's effects as well as the temporal windows of their necessity/sufficiency. For example, the authors can conduct cell-specific rescue/depletion experiments of cat-2/dop-2/dop-3 to address the spatial requirement/necessary circuit for dopamine, while supplementation or AID (auxin-induced degron) experiments can address the temporal requirement of dopamine/serotonin. These results will provide deeper mechanistic insights and a better understanding of how these neurotransmitter systems contribute to the effects of early-life starvation on roaming behavior.

We agree with the reviewer that further dissection of the exact circuits involved and their temporal requirements for affecting stage-specific and individual variation will provide deeper mechanistic insight. While this is beyond the scope of the current paper, we will definitely pursue these interesting directions in future studies.

2) The study largely focuses on a single aspect of behavior: roaming vs. dwelling. How early starvation affects this one behavior parameter and how dopamine/serotonin play a role in it may not be broadly applicable to other behavioral parameters. This was even demonstrated in the manuscript itself, as the authors also looked another related behavioral parameter, speed during roaming, and found different effects of early-life starvation and of dopamine/serotonin on this parameter compared to their effects of average roaming fraction (most prominently demonstrated in Figures S4D vs. C and S7C vs B, but also in Figures S1E vs D and S3C vs. B). The authors are using a very unique technique to be able to capture the entirety of the animal's development, and there are likely many other parameters that can be extracted from this valuable resource. The examination of many different parameters will lead to a better overall picture of how behavior is influenced by early life experiences.

As the reviewer noted, the multiple behavioral effects discovered using the two behavioral parameters that were quantified in this study implies that many other parameters that can be extracted from our dataset will potentially uncover more behavioral effects. In fact, in the discussion we clearly note that the behavioral parameters that we quantified in this study represent only a subset of the behavioral repertoire available to the organism. While quantifying how the full behavioral space across development may be modified under different experiences and neuromodulatory states is not the current focus of this study, it is a major long-term theoretical/computational research direction in our lab.

3) There are some confounding factors/technical limitations that are not considered or clearly presented in the manuscript that may alter overall interpretation of the data/conclusions.a. It is unclear from the current presentation of data that the authors have sufficient resolution to accurately calculate the behavioral parameters of younger, smaller worms, particularly those at the L1 stage. This was evident in the authors' own data that cat-2 mutation did not increase the speed of the L1 worms but at all other stages, and could be as a result of decreased sensitivity/resolution at this stage. The authors should provide evidence that their analysis provides sufficient resolution to measure behavioral features of L1 animals by comparing their analysis/results to more high resolution approaches to validate some of their results. For example, control and cat-2 mutant L1 in their approach vs. a higher resolution approach.

The reviewer correctly states that lower spatial resolution in younger L1 worms may potentially limit the ability to detect small behavioral changes. However, using our imaging system, we were able to extract multiple behavioral effects in L1 individuals, such as an increase in roaming activity and roaming speed during L1 in *tph-1* and *npr-1* mutants, respectively (this paper and Stern et al. 2017), and a decrease in roaming activity during L1 following starvation (this paper). However, we agree with the reviewer that it is possible that milder/smaller behavioral effects may be harder to detect. We have now added a statement about this spatial resolution limitation in the discussion (p. 16).

b. Somewhat related to a, the animals change dramatically in size from L1 to the adult stage. This was not taken into account of the calculation of roaming vs. dwelling behavior. Does altered size affect roaming vs dwelling behavior? If so, does starvation and dopamine/serotonin affect size of the animals at different stages, and does this fully or partially explain their effects on roaming/dwelling behavior? The authors should be able to extract length/width data from their recordings and address these questions. This would significantly affect the interpretation of the current manuscript.

For quantifying roaming vs. dwelling episodes during different developmental stages in which animals have different sizes we used a calibrated parameter of roaming/dwelling threshold that is specific to each developmental stage (methods section of this paper and of Stern et al. 2017) and that we found to robustly define roaming episodes across different developmental stages. In addition, in the revised manuscript we include new analyses of size and roaming/speed data in single animals (based on individual size extracted from images) and compare behavior across size-matched individuals (within the same size window) across conditions/genotypes. In particular, these analyses show that while the average size in the starved populations is slightly decreased (~10%), size matched individuals across conditions/genotypes (within the same size range) show similar stage-specific behavioral effects to the ones shown using a comparison of the whole population (Figure 1 —figure supplement 1F; Figure 3 —figure supplement 1D; Figure 5 —figure supplement 1D). Furthermore, these analyses demonstrate, as previously shown, an increase in roaming fraction in sized-matched *tph-1* individuals compared to wild-type (Figure 5 —figure supplement 1B), as well as an increase in roaming speed in the L2-Adult stages in size-matched *cat-2* individuals compared to wild-type (Figure 3 —figure supplement 1E).

4) By the title and abstract of the manuscript, the authors promises to answer the question of how early life starvation affects individuality. However, there is no clear presentation/conclusion in the text/figures of how early life stress affect individuality. For example, are there more/less variation across different principal components after early-life starvation as shown in Figure 2/S2? Rather the authors focused on how dopamine/serotonin affect individually, which was already previous reported in a previous manuscript (Stern et al., 2017). The authors already have many of these analyses presented in the current manuscript (Figure 2, 6, S2, S8), but need to directly compare whether starvation affects variation/individuality in roaming behavior.

We thank the reviewer for this comment. As the reviewer suggested, in the revised manuscript we include a systematic and direct comparison of inter-individual variation between stressed and unstressed populations across multiple PC individuality dimensions. These analyses directly show modifications in inter-individual variation in specific PC individuality dimensions (PC1, PC3, PC6) following early-life stress, within wild-type and neuromodulatory mutant populations (Figure 6; Figure 6 —figure supplement 1). In addition, we also made textual edits (abstract and main text, p. 11-12) to provide clearer description of the effects of early-life stress on variation in specific PC dimensions.